# Research Animal Behavioral Management Programs for the 21st Century

**DOI:** 10.3390/ani13121919

**Published:** 2023-06-08

**Authors:** Patricia V. Turner, Kathryn Bayne

**Affiliations:** 1Global Animal Welfare & Training, Charles River, Wilmington, MA 01887, USA; 2Department of Pathobiology, University of Guelph, Guelph, ON N1G 2W1, Canada; 3AAALAC International, Frederick, MD 21703, USA; kbayne@aaalac.org

**Keywords:** laboratory animals, enrichment, animal welfare, animal behavior management, animal welfare assessments

## Abstract

**Simple Summary:**

Continued evolution of how research animals are kept and cared for is vital for respecting the 3Rs of replacement, reduction, and refinement. How animals feel forms the basis for their behavior and for determining their welfare on a day-to-day basis. We propose broad use of an updated behavioral management program approach that emphasizes considering the full range of research animal needs and the outcomes that are desired when manipulating their environment. Use of this approach will improve animal welfare but also the quality of the science obtained from working with these animals.

**Abstract:**

Behavioral management programs have been developed commonly for research dogs and primates but rarely has program consideration been expanded to include all research species worked with. This is necessary to reduce animal stress and promote natural behaviors, which can promote good animal welfare and result in more robust and reproducible scientific data. We describe the evolution of consideration for research animal needs and define an umbrella-based model of research animal behavioral management programs, which may be used for all research species. In addition to developing a more comprehensive program, we emphasize the need for regular welfare assessments to determine whether the program is working cohesively and whether any aspects require modification.

## 1. Introduction

Behavioral management programs are not a new concept for research animals but considerations for implementing them, at least in North America, have largely been limited to a few species of higher public interest, such as dogs and primates. As our understanding of animal sentience has evolved [1,2,3] and many in the field have embraced a care-based ethical approach [4,5], the research community has shifted to emphasize positive animal welfare states [6,7,8] and animal welfare assessments have expanded to become more wide-ranging, employing an updated Five Domains assessment model [9], rather than focusing primarily on physical health and minimizing pain and distress. Through this evolution of research animal ethics, assessment of care, and animal welfare science, gaps in our approach to care for some species, such as mice, rats, and pigs, have become evident. While the basics of care are provided for all species, there has been less emphasis on optimizing engagement of animals more fully with their environment and with those who work with them in research settings, to optimize their welfare. In other sectors of animal care and husbandry in which animals are managed for extended periods of time, such as Association of Zoos and Aquariums (AZA)-accredited zoos and aquaria, significant consideration is now given to managing the behavior of all species, including rodents, birds, and fish [10,11,12]. This is largely accomplished through development and implementation of species-specific, outcomes-based behavior management programs (BMP) [13].

In addition to ethical concerns to ensure good animal welfare, there are both regulatory and scientific imperatives for adopting behavior management programs for research animals. The EU Directive 2010/63 [14] has long emphasized behavior management and training of animals as part of refinement of animal care (although initially training support was geared to large animal species). In other sectors, such as food animals, periodic animal welfare assessments have long been discussed and described as a means of ensuring responsible care of animals and addressing public accountability [15,16,17]. In Canada, the Canadian Council on Animal Care (CCAC), also promotes regular animal welfare assessments for all species [18] as well as the need for institutions to expand their approach to behavioral management of research animals [19,20,21]. As part of the scientific community’s consideration of reproducibility of scientific work [22,23], there is heightened awareness that adverse stress impacting an animal can arise from experimental procedures, but also from insufficient attention to the ways in which animals are managed on a day-to-day basis [8,24]. While aspects of expanding animal management programs to better account for behavioral needs may add some experimental variability, the emphasis on working with normally behaving animals also means that greater confidence can be placed in results obtained from research with these animals [25,26]. This need exists whether animals are in the vivarium for five days or five months [18]. This paper will further explore concepts of research animal enrichment and behavior management—both from a historical approach as well as defining what is meant by research animal behavioral management, with a suggested behavior management model approach, as well as covering assessment of behavioral program impact, and areas of future research needs.

## 2. Conceptual Development of Environmental Enrichment and Animal Behavioral Management

Early efforts aimed at improving the environments of captive animals began with the zoo community. Mellen and Sevenich MacPhee [27] trace the recognition of the value of enrichment back to Robert Yerkes in his publication *Almost Human* [28]. Hediger [29] expanded the concept of enrichment for captive animals, noting both the biological and psychological impacts of confined space. He shifted the traditional thought paradigm from a narrow concern about the limited freedom of movement for captive animals to the overall quality of their environment, stating that “the quality of the space … is of the greatest importance for its [the animal’s] welfare.” He argued that anthropomorphism must be abandoned in favor of putting ourselves in the animal’s position to understand its needs [29].

Regrettably, despite the admonitions of these early advocates of improved captive animal welfare, barren environments and the resulting reduction in welfare, evidenced in part by very obvious stereotypic behaviors manifested by animals of diverse species, characterized the housing of zoo animals for decades. The intended benefits of naturalizing the animal’s environment immediately centered on the improved welfare of the animal and the consequent enhancement of conservation efforts. The strategies used to increase the expression of species-typical behaviors in captive zoo animals encompassed reproductive and maternal behaviors [30], which would promote species conservation, but were also directed at a variety of other behaviors that the animal would manifest in a naturalistic environment (e.g., nest building, social interactions, foraging for food, etc.). As a result, an additional advantage to enhancing the environments of zoo animals was the educational messaging provided the viewing public regarding the highly complex and rich lives of these animals.

The provision of environmental resources to animals worked with in science grew out of an increasing recognition by scientists and veterinarians that the improvements in the welfare of zoo animals through intentional design of their living environments could, and should, be extrapolated to other captive environments, such as the research facility [31,32]. These thought leaders promulgated the idea that the quality of the research data generated from research animals is dependent on the overall quality of life of the animals, not just the absence of disease in the animals (e.g., [33]). However, consideration of ways to expand an animal’s behavioral repertoire in the research setting to better reflect the range of species-typical behaviors initially appeared to be in direct contradiction to practices that had been in place for decades: the perceived need for an environment that could easily be sanitized to minimize infections in research animals; the perceived need for consistency in how animals were housed to minimize the introduction of potential confounding variables into the research data; and minimizing risks to the safety of the animals and the personnel [31,34].

Other barriers to providing resources that were proposed included the cost (the actual expense of items, how this would be funded, personnel time in distributing resources), the potential for the resources to compromise the physical health of animals (e.g., treats that might lead to obesity; foreign body obstructions resulting from animals chewing on and swallowing parts of enrichment devices, entrapment of animals, etc.), and discerning whether providing resources resulted in an actual benefit to an animal or was simply an activity that the staff believed was helping—in the absence of any objective evidence. In many ways, the most significant impediment to including resources was that large bodies of data (i.e., historical control data) had been collected from animals living in a sterile environment and there was widespread concern among scientists that modifying that environment would also alter the subsequent data [35], although the counter-argument has also been made (see for example, [36,37]).

A further early complicating factor to the broad acceptance of the implementation of environmental resources was the numerous ways in which it was defined. There was an early inclination that continues to the present to define enrichment in the laboratory as additions to the animal’s living environment, such as a toy or food treat, without consideration of the gestalt of the animal’s well-being. For example, a hard ball was provided to nonhuman primates and described as an enrichment [38]. Yet, often these so-called enrichments were found to be of limited lasting interest to the animals [34,39], and thus their value in truly improving the welfare of the animals was challenged. After the approach of providing simplistic objects to the animal’s enclosure was repeatedly challenged as having a positive welfare effect, consensus has generally evolved to consider the entirety of the animal. Specifically, enrichment in the zoo setting was accepted as “… an animal husbandry principle that seeks to enhance the quality of captive animal care by identifying and providing the environmental stimuli necessary for optimal psychological and physiological well-being” [40].

For research animals, definitions of environmental enrichment continued to serve more as a barrier to progress than as guidance to improving animal welfare. As enrichment was making its way into the laboratory environment, Chamove and Anderson [41] suggested an approach that could be objectively measured, namely that enrichment should decrease “undesirable” behaviors and increase “desirable” behaviors. While measurable, the definitions of desirable and undesirable behaviors for a captive animals could be imprecise, as they could be shaped by the specific context of behavioral expression, its frequency and intensity, and possibly observer bias. Similarly, enrichment has been characterized as an environment that fosters “natural” behavior (e.g., [42]), though as Newberry [43] has noted, what is “natural” may be equally difficult to define, as the benchmark may be behaviors expressed in the wild, in a spacious outdoor environment, or an environment that includes elements of the natural habitat of the animal. As time has passed, the definition of enrichment shifted to focus on the goals of the enrichment (though there remain some consistencies in the definition over time, see Table 1), such as “an improvement in biological functioning” [43] or providing more choices to an animal as well as more control over its environment [43,44]. These approaches have become the foundation upon which other goals have been added, such as expanding the range of behaviors expressed by the animal, reducing the expression of abnormal behaviors, increasing “positive utilization of the environment,” and improving an animal’s ability to cope with challenges in its environment [45].

Several factors converged to place nonhuman primates at the forefront of the provision of environmental enrichment to research animals. Novak and Petto [46] noted: “It is easy for us, as humans, to see ourselves reflected in the lives of our nonhuman primate cousins.” Arluke and Sanders described a sociozoologic scale [47] based on societal ranking of animals and a “ladder of worth.” Societal concern and empathy for the welfare of nonhuman primates led to the 1985 amendments to the Animal Welfare Act [48] that require “a physical environment adequate to promote the psychological well-being of primates.” Implementing regulations subsequently published by the U.S. Department of Agriculture (USDA) in 1991 to address the requirement that “dealers, exhibitors, and research facilities must develop, document, and follow an appropriate plan for environmental enhancement adequate to promote the psychological well-being of nonhuman primates” [49]. As a result, that is where research institutions initially focused their efforts. Another potential consequence of the terminology used in the AWAR is that the term ‘environmental enrichment’ (a.k.a. enhancement) became embedded in the U.S. lexicon, thereby possibly limiting earlier development of a broader concept of the research animal’s overall life experience. Publications related to environmental enrichment of animals in different environments (zoo, research, agriculture) were modest in number prior to 1991, but have been trending upward since then [50] as enhancement strategies have been investigated for a wider variety of species, in part due a natural extension of the benefits of improving environments for all the animals used in research but also due to the requirement for enrichment to be provided to all research animals required by various regulatory frameworks (e.g., the European Directive [14]).

Further amplifying the initial prioritization of enhancing the environments of larger research animals, such as dogs and primates, was the greater general familiarity with them, and the resulting knowledge base upon which to initiate lines of investigation for impactful enrichment strategies. These are also species of high interest to the general public (i.e., there is a cultural sociozoologic bias for these species). The behavior of dogs and primates is more extant, better understood, and easier to assess than that of smaller research animals such as mice. Thus, effective improvements to the environment could more readily be correlated to improvements in the welfare of these species. However, over time, the provision of an improved environment has rippled out to other species of research animals, and research into methods of improving and assessing the welfare of small mammalian research species, such as mice, rats, and rabbits, has dominated the recent enrichment literature.

As studies on environmental enrichment and the welfare of laboratory animals matured, the limitations of a narrow perspective on enrichment became quite evident. The impact of simplistic modifications or additions to the research animal’s environment, while beneficial if well considered, did not necessarily result in an improved life experience for the research animal in all aspects of its day. The concept of a behavioral management program evolved incrementally out of the variety of approaches described to enrich the research animal’s environment. An early indicator that acceptance of simplistic methods of providing enrichment (e.g., adding a toy to the cage) was being challenged was illustrated by the creation of the National Institutes of Health (NIH) Management Plan for Nonhuman Primates [51]. This plan articulated the value of social housing, the need to involve scientists in approaches to enhance laboratory primate welfare, the need to establish methods of assessment, offered suggestions to improve both the primary and secondary enclosures of the animals, addressed the impact of husbandry routines, the value of offering a varied diet to some species, and noted the value of training animals to cooperate in procedures. This early description of a more holistic approach to improving the research animal’s welfare has subsequently developed into a “behavioral management program” (BMP). The BMP has been defined as “a comprehensive approach to improving the welfare of captive animals by employing social housing, environmental enrichment, animal training, facility design, and the assessment of behavior and behavioral problems” [52] and, conceptually, has been promoted by numerous organizations, such as the Animal Behavior Management Alliance, the Association of Zoos and Aquariums, the Universities Federation for Animal Welfare, and others. Depending on the species and manner of housing, a BMP may be applied to an individual animal or socially housed animals. In general, programs are developed for a species and then tailored to the individual, when possible.

**Table 1 animals-13-01919-t001:** Examples of additional definitions of ‘environmental enrichment’ in terms of methodologies and goals.

“additions to an animal’s environment with which it can interact”	[53]
“ameliorate problems caused by containment”; “alter behavior so that it is within the range of the animals’ normal behavior”	[54]
increase species-typical activities, reduce behavioral pathology, provide social interaction, increasing the diversity of foods offered, improve the quality of interactions between people (animal care staff, investigators, veterinarians) and the animals	[55]
“species-adequate stimulation”	[56]
“stability and security”; “opportunities to achieve goals”, “complexity and unpredictability”	[57]
providing animals with “broad contingencies and allowing them to ‘invent’ ways to use them”; “control over the environment”	[58,59]
social companions and “stimulus complexity”	[60]
refers to modifications that act to enhance the level of physical and social stimulation provided by the captive environment	[61]
“provide opportunities for the animals to perform species-specific behavioral repertoire”	[62]
“allow for more exercise, play, and compatible social interaction”	[63]
“to improve the welfare of the animal”	[64]
“generally refers to items we provide to the animals to support their behavioral needs. It provides a way to functionally simulate the natural environment of captive animals, in an effort to increase opportunities for the expression of species-specific behaviors and decrease the occurrence of abnormal behaviors.”	[65]
“a housing condition in which animals benefit from the sensory, physical, cognitive and social stimulation”	[66]

## 3. Developing a Holistic Model for Research Animal Behavioral Management

### 3.1. Establishing the Need for Better Management of Research Animal Behavior

As discussed, research animal behavioral management refers to an essential program for ensuring the health and welfare of animals worked with in science and education. Behavioral management programs should employ a comprehensive approach, encompassing thoughtful consideration of the significant factors that may impact a research animal’s behavior and welfare [6,52,67]. There is an ethical concern for the types of experiences that research animals have, with particular interest in the subjective experience of animals and their capacity to suffer [68]. Although ensuring the absence of disease is an important consideration in the assessment of animal welfare, examining animal behavior goes one step further and includes observing an animal’s emotional state. Observation of and reflection on animal behavior provides insight into animal motivations and preferences, and has become the epicenter of animal welfare assessment [9]. Behavioral management programs use behavior as the central construct for examining numerous variables that impact research animals’ lives.

Behavior management programs have the potential for creating a powerful and positive impact on animal lives, and may also help to improve the quality of scientific research. The quality of data collected is thought to be directly related to the welfare of the animals involved in a study [25,69]. Animal management factors including the housing environment, husbandry procedures, and handling/restraint techniques used, directly impact animal physiology and welfare, and thus the research data. For example, a recent systematic review of 165 peer-reviewed scientific papers demonstrated that mice housed in conventional, minimally stimulating environments that are common in many research facilities around the world are more susceptible to morbidity and mortality when diseases or conditions such as stroke, cancer, and anxiety are induced [25]. Similarly, epigenetic changes in neuronal activity and associated improvements in cognitive deficits are seen in adult female Alzheimer’s disease-susceptible (fxFAD) mice when they are moved to an environment with improved resources such as tunnels and huts [70]. Adverse mouse handling techniques (i.e., tail handling) has been demonstrated to impact metabolic research, shifting blood glucose and corticosterone curves, and confounding interpretation of research results [71]. Similar shifts in blood glucose levels are seen in rhesus and cynomolgus macaques that are not habituated to handling prior to blood sampling [72]. Use of non-aversive handling techniques (i.e., tunnel-handling) for breeding mouse colonies has been associated with increased pups born and weaned per litter, important for both breeding colony management as well as reproductive studies examining fertility [73]. As a final example, provision of adequate nesting material for mice has been demonstrated to improve mouse breeding performance and is recommended for thermoregulation of mice to enhance translatability of mouse-based molecular research [74,75]. These examples demonstrate that implementation of robust behavioral management programs have the potential for improving not only research animal welfare, but also data quality and reproducibility.

### 3.2. Conceptualizing an Umbrella Model for Research Animal Behavioral Management Programs

Umbrellas have long been used as a symbol of protection and shelter, and in recent years, an ‘umbrella concept’ has come to signify a broad or overarching range of concepts or terms belonging to a common category [76]. For example, an umbrella model has been used to depict both the One Health and the One Welfare concepts to symbolize an all-encompassing model with clusters of related factors depicted as being sheltered by the umbrella [77,78]. An umbrella model is similarly proposed for research animal behavioral management programs (Figure 1) to highlight the related variables, derived from more than two decades of literature, that impact research animal welfare and behavior for a given species in a graphical format. The umbrella as a concept is used intentionally to convey the importance of clustering and considering how all of the components of an animal’s environment may change, suppress, and give rise to various behaviors, including abnormal ones. Using an umbrella to depict this model also emphasizes that program elements should be protected together as a whole when developing the behavior management program, instead of considering them as individual items to select from, a more common approach when developing ‘enrichment programs’.

The raindrops on the umbrella in this model represent the impacts for creating and maintaining a robust research animal behavior management program. This includes the costs for adequate resources and personnel to implement the program; the availability of equipment and adequacy of facilities; the time required to manage animals; challenges presented by tradition (e.g., a holistic approach is less common for species other than primates and dogs); public expectations and transparency about how research animals live and are cared for; researcher or client expectations for research quality; the regulatory environment under which research animals are kept and maintained; validation of program elements related to how animals are managed; compassion stress and fatigue of personnel, which can be reduced with thoughtful resource management programs for animals [79]; and institutional employee satisfaction and retention.

For this model of research animal behavioral management programs, 12 components have been identified as being essential elements for consideration, and include items to chew and manipulate, food novelty or opportunities for foraging, behavioral assessments, social housing, animal comfort, human interactions, housing and husbandry, handling and restraint, habituation and desensitization, exercise, choice and control in the environment, and animal training. The rationale for selecting these elements is that they have all been individually identified as being highly important for enhancing the welfare of multiple research animal species (for reviews, see [24,62,80,81,82,83,84,85]). For specific instances in which one component or another has not been specifically investigated in a given species, a ‘bias for action’ approach can be considered [86]. An essential consideration for the model is that although 12 individual elements are listed separately, often by making a change in one program element, other elements may be impacted. For example, addition of an appropriately sized tunnel to a mouse or rat cage not only provides a sheltered and darkened place for resting (housing and husbandry; comfort), but it also compartmentalizes the cage (choice and control), inserts an object that can be climbed upon or around (exercise), provides an object that can be chewed (things to chew and manipulate), and can be used for nonaversive handling (handling and restraint).

### 3.3. Emphasizing an Outcomes-Based Approach for Research Animal Behavioral Management Programs

When considering each of the variables under the umbrella an outcome-focused approach should be used for optimizing animal care [13]. Outcome-focused approaches for animal management consider first what are the types or behaviors, postures, experiences, and activities that are normally expressed by a given species and then considers how the essential elements of these aspects may be recapitulated in a captive setting [13]. That is, when any factor under the umbrella is being considered, one should ask what specific outcomes in animal behavior and well-being are desired or will be addressed [80]? This approach will also help in determining whether program modifications have been successful. Examples by species of different elements of the behavioral management program are provided in Table 2. As outlined, this should include consideration of environmental resources or human or animal interactions that provide research animals with the opportunity to perform inherently motivated species-specific behaviors, postures, and other activities. One must also integrate consideration of animal housing and comfort, social interactions and buffering [87], providing elements of choice and control for all species [88], and better preparation of animals for experiments through habituation, and training (for example, [89]. A large portion of an animal’s time is spent in their housing environment outside of experimental activity, and this overall environment has a significant influence on an animal’s comfort, behavior, and overall welfare. It is important to note that even the most thoughtful RABM will not recapitulate the natural, free-living state of an animal. However, this approach ensures that many animal needs are met, that adverse stress is reduced, and it sets the stage for positive and cooperative interactions between people and animals in research settings.

### 3.4. Responsibility for Developing Research Animal Behavioral Management Programs

Development of a comprehensive behavioral management program for each species is a joint responsibility that may need to involve multiple stakeholders, including the IACUC or Animal Ethics Committee, affected researchers and their teams, husbandry personnel and their supervisors, veterinary professionals, and research animal behaviorists or behavior champions. Certainly, for the RABM to be sustainable in the long term it must consider the needs, questions, and abilities of animal facility personnel. The ability to change the approach to research animal management will be highly dependent on collaboration and consensus-building as well as the underlying institutional culture. For some species and institutions, this approach may be largely in line with ‘enrichment programs’ that are already well established, with minor adjustments needed to ensure an outcome-based focus. For other institutions or species, a behavioral management program approach may require a significant shift in thinking and investment in equipment, caging, training of facility personnel and research groups, and other resources. Piloting and validating program approaches with a small subset of animals can help to achieve stakeholder buy-in and justify expenditures. While a needs assessment and development of a behavior management program can be enhanced or accelerated with the assistance of a trained research animal behaviorist or behavior consultant, many excellent programs have been developed by self-taught individuals with a strong interest in animal behavior and access to animal behavior consultants and/or peer-reviewed behavior literature.

In the process of developing a program, setbacks may occur and it is important to document, discuss, and share these challenges. New facility modifications need to be well validated before being widely implemented to ensure the safety of both people and animals. It is also important to be flexible and to adapt programs to research specifics. For example, implementation of a clicker training program (operant conditioning) requires knowledge of the technique, consistency in implementation, time, and practice [117]. It is successfully used by many lay companion animal owners on a daily basis and can also be used effectively for training most domestic (and many nondomestic) animals [118]. However, this technique may not be the best choice for training of animals on short acute studies, studies in which animals will be infrequently worked with or handled, or there is a lack of consistency in personnel working with the same experimental animals each day and/or a lack of understanding of how to correctly employ the technique by all personnel involved in handling animals. For short, acute studies time might be better spent on desensitizing animals to caregivers and encouraging positive human–animal interactions [79].

## 4. Monitoring the Impact of Research Animal Behavior Management Programs

Improving the care and management of research animals is an ongoing process that requires periodic review and assessment. This is a requirement in some countries, such as the UK (Animals (Scientific Procedures) Act, 2012) [119], Canada [18], and in the EU [14], and it is an important activity whenever animals are maintained for human purposes to ensure that welfare needs are being met and addressed [120]. Further, because of financial constraints and resource requirements, program enhancements may need to be incremental and it is vital to assess these to ensure that resources are well placed.

Monitoring the impact of the behavior management program goes beyond completing a simple checklist that asks whether a particular resource is present or not for every animal during a room walkthrough. Instead, the assessment should determine how animals are behaving and interacting with their environment, social partners, and with humans—both familiar and unfamiliar. For this reason, it may be beneficial to have individuals more familiar with a given species or group of animals conduct the assessments, such as an in-room caregiver or research technician or behaviorist.

For some species, such as mice, monitoring individual animal or even small groups of animal interactions with their environment may not be feasible if very large numbers are held at the institution. In such cases, the general welfare of animals may be assessed by a surrogate measure of well-being, such as a change in the cage that can be determined quickly and across large numbers of cages, for example, the time-to-integrate-to-nest test (TINT; [121]). Using this assessment tool, integration of new nesting material into the main nest by mice is judged to indicate an animal is not in pain following a procedure as their state of welfare supports the expression of a common normal behavior. Even more simply, the quality of a nest and whether and how mice are using it during their resting phase may be scored to determine adequacy of resources provided, in-cage group dynamics, and overall welfare of mice [122,123]. A similar challenge is encountered with zebrafish used in research. The enrichment of fish tanks is undergoing more systematic evaluation (see, for example, [124]), and new tools are in use to assess the animals’ behavior, such as the use of video recordings with computer software that relies on a behavioral evaluation algorithm [125].

Regardless of the assessment tools used, single-point assessments are inadequate. Rather, an accurate determination of the impact of the behavior management program can only be made if multiple assessments are conducted over time. For example, animals may lose interest in some forms of resources provided and return to exhibiting abnormal behaviors; social interactions may evolve as animals age, which could result in incidents of conflict that require other adjustments; or other changes in external factors (e.g., changes in personnel, change in animal composition in the primary enclosure or room) may have an effect on the welfare of the animal [126]. The frequency and type of assessment should be based on the species of animal, type of research being conducted on the animal, and its baseline behaviors, among other factors [18]. Even when the same groups of animals cannot be followed over time, because of the nature of the research being conducted, regular and repeated species-specific assessments of animal welfare are valuable as they may demonstrate trends that need to be addressed. Examples may include an increase in bites or scratches, changes in housing- or equipment-related injuries, etc. Changes detected in the impact of the behavior management program on animal welfare should be shared with the veterinary team, the research group, and potentially the IACUC/AEC to determine program needs, scientific impact, and future directions.

Although introduction of more formal assessment programs initially creates an extra burden for a workforce that is already often taxed, this also represents an opportunity for facility husbandry and technical personnel to actively engage in animal management decisions and to better understand the needs of the animals they work with daily. This empowerment has been demonstrated to be important for enhancing compassion satisfaction for those working with research animals [127].

## 5. Areas of Future Research

Given the breadth and comprehensive nature of behavioral management programs, there are many areas for which research is needed to provide evidence-based recommendations for these programs. This includes seeking ways to enhance opportunities for human–animal interactions, particularly looking at when more intensive interactions may be beneficial, and how this may be done efficiently with large numbers of animals. Other investigations will need to consider how to provide more optimal housing for a range of species, including mice and rats, and ways to provide exercise opportunities for a range of species, again, including rodents. This research is inherently applied and must consider species’ biology and behavioral responses to alterations, as strategies for improving research animal welfare may not necessarily be efficacious for improving welfare under many different settings and conditions. For example, one strategy may work well with one strain of mouse in a particular housing environment, and may not necessarily translate as beneficial for another mouse strain under a different set of environmental conditions [37,86]. Given the diversity of strains, species, and research settings, it is important to consider that various strategies for improving research behavioral management, and ultimately animal welfare, will be required. Much of this will come down to the willingness of institutions and researchers to incorporate comprehensive behavioral management programs into their experimental protocols and research personnel schedules.

It is also important to consider change management and practicality. For strategies to be implemented and improve research animal welfare, it is important they are feasible. Feasibility includes considering the economics of implementing the strategy, the impact on the research and animal care professionals, as well as the impact on the science these animals are bred to take part in.

## 6. Conclusions

Progress in improving care and management of research animals requires a more comprehensive approach. Use of the proposed ‘umbrella model’ to construct a behavioral management program for each species worked with will ensure an integrated approach that helps researchers and caregivers to focus on outcomes for providing suitable housing, resources, and other elements necessary for animal well-being. Use of this plan in conjunction with periodic welfare assessments will help animal ethics committees, researchers, research animal professionals, and research institutions to ensure that their programs remain up-to-date and that ongoing attention is given to continuous improvement of the program. In addition to expected improvement in research animal welfare, increased attention to behavioral management is expected to enhance research outcomes, by ensuring that experimental animals are behaving and functioning normally. Behavioral management programs may also improve compassion satisfaction for those directly caring for and working with research animals, an important consideration for mental health and well-being in the workplace and long term employee retention.

## Figures and Tables

**Figure 1 animals-13-01919-f001:**
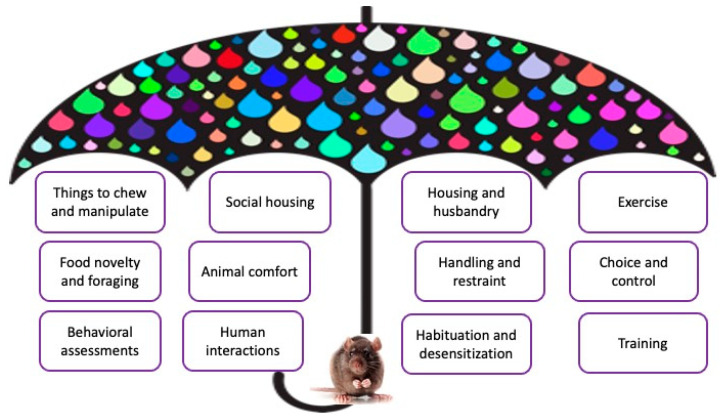
A proposed umbrella model of research animal behavioral management programs. A rat is depicted for this image but can be replaced when considering other research species.

**Table 2 animals-13-01919-t002:** Examples of elements of a outcomes-focused research animal behavioral management program (RABMP) for several common research species.

RABMP Component	Description and Outcome	Species-Specific Examples				
		Macaque	Dog	Rat	Rabbit	Pig
Manipulanda or objects to chew	Objects used to reduce boredom, stimulate cognitive function, and occupy time; may also be chewed or rooted as part of sensory experience or to prevent dental overgrowth for species with continuously growing teeth (e.g., rodents and rabbits); used when desired by animals [90]	Kong toys, balls, rings, manzanita wood [91]	Kong toys, balls, bones	Nylon bones, or sticks, cardboard tubes or objects, aspen wood sticks [92]	Plastic bottles, nylon toys, metal chains, aspen wood blocks [92], empty paper feed or bedding bags	Boomer balls, balls in kiddie pool, empty paper feed bags, rubber ‘jacks’
Social housing	Consideration for ensuring that social species are housed and maintained socially—outcome being to minimize fear, anxiety, and stress, and to promote normal species-typical behaviors, social buffering [87], comfort, and psychological health [93]. When not possible to continuously socially co-house, consider other options, such as allowing co-housing at night, use of grids between enclosures, use of Plexiglas or glass between enclosures, co-housing unpaired study animal with naïve animal, etc.	House with one or more socially compatible con- specifics. Visual, olfactory, and auditory contact, when not possible	House with one or more socially compatible conspecific. Visual, olfactory, and auditory contact, when not possible	House with one or more socially compatible conspecific. Use of larger cages with fixed plastic dividers when social housing is not possible. Visual, olfactory, and auditory contact, when not possible	House with one or more socially com- patible conspecific. Visual, olfactory, and auditory contact, when not possible	House with one or more socially compatible conspecific. Visual, olfactory, and auditory contact, when not possible
Food novelty and foraging	Most species worked with spend many hours each day in natural habitats searching for food. Outcomes: provides dietary variation and trace nutrients, promotes sensory and cognitive stimulation (e.g., puzzle feeders), promotes gross and fine motor skills coordination, helps to occupy animals during day, can be used as reward during training. Food supplements should be cut to an appropriate size and not exceed >5% of the daily diet to avoid unbalancing the diet and contributing to obesity	Puzzle feeders [94,95], foraging boards, popcorn (popped in the room with a hot air popper), providing food in a different manner (e.g., in a bag or box, on top of enclosure), fresh fruits and vegetables, commercially available primate treats, seeds or foods sprinkled in clean substrate for foraging [96]	Commercial dog biscuits and treats, fresh fruit and vegetables, peanut butter or frozen applesauce/treats in Kong toys, popcorn	Toasted oat cereal, fresh fruit and vegetables, raisins, dried cranberries/fruit, seed mix sprinkled in bedding, popped popcorn	Fresh fruit and vegetables, use of cardboard tubes to create puzzle feeder, commercial rabbit treats, hay	Seeds, raisins or trail mix in kiddie pool with water or balls, fresh fruit and vegetables, cereal, straw (rooting)
Comfort	In the search for hygienic enclosures that are safe and readily sanitized, comfort is sometimes forgotten; refers to surfaces animals are in direct contact with as well as elements of facility design and enclosure features to reduce fear or stress [97]. Outcomes: better quality rest and sleep, decreased health issues (e.g., interdigital cysts, pododermatitis, abrasions, hoof fractures, etc.), and decreased fear and stress	High resting places, multiple perches, use of thermoneutral materials for resting surfaces (e.g., wood or plastic), use of substrate in enclosure bottom, visual ‘hides’, use of horizontal bars on gates and doors to improve in-room visibility	Solid floor housing, elevated resting area (thermoneutral), use of substrate, use of beds	Solid bottom cages, provision of preferred substrate (e.g., wood or cellulose chip), nesting material (e.g., shredded or crinkle paper), tunnel or shelter	Solid floor housing with substrate, elevated resting area, area for hiding (tube, large paper bag, shelter or space under platform), housed away from noisy species such as dogs	Solid non-slip floor housing, use of substrate, plastic baby baths (mini-pigs only), use of horizontal bars on gates and doors to improve in-room visibility, visual hides for privacy, nesting materials, e.g., Excelsior, straw
Behavioral assessments	Used to assess animal compatibility with conspecifics as well as to evaluate presence of abnormal behaviors, use of resources, temperament testing, etc. Outcome: well adjusted, normally behaving animals that are provided with safe, interesting and appropriate resources. Depending on institutional size, animal population and duration of stay, and nature of ongoing research, may include repeated assessments of the same or different animals over time. Assessments and any necessary actions should be documented [18]	Periodic and ‘for cause’ (when abnormalities are noted) behavior assessments	Periodic and ‘for cause’ (when abnormalities are noted) behavior assessments	Periodic and ‘for cause’ (when abnormalities are noted) behavior assessments	Periodic and ‘for cause’ (when abnormalities are noted) behavior assessments	Periodic and ‘for cause’ (when abnormalities are noted) behavior assessments
Human interactions	One or more daily positive human–animal interactions. Outcomes: reduction in animal fear, desensitize to the presence of people and promotion of trust, pleasure and relaxation, as well as enhancing human attentiveness, engagement, and satisfaction when observing and working around animals [98]	Provision of small food treats, quietly talking or singing to animals, reading children’s board books to animals and showing them pictures	Provision of small food treats, play time, leash walking, quietly talking or singing to animals, reading children’s board books to animals and showing them pictures	Provision of small food treats, petting	Provision of small food treats, grooming, petting, talking softly	Provision of small food treats, play time, grooming, quietly talking or singing to animals, reading children’s board books to animals and showing them pictures
Housing and husbandry	Provision of housing that meets species-specific behavioral needs, including adequate space for normal postural adjustments and locomotory actions, such as stretching vertically and horizontally, jumping, etc. Also includes facility design elements that contribute to appropriate housing and environment. Husbandry management techniques that reduce fear and distress in animals, e.g., quiet mannerisms when conducting husbandry, nest transfers for rodents, ensuring animals are moved and not wetted during cage or enclosure cleaning, etc.	Perches or shelves at different heights and made of thermoneutral materials; enclosures that emphasize vertical height; horizontal bars on fronts of enclosures; use of balconies or tunnels that enhance vantage of the room; use of daylight bulbs and natural light, such as through a window or skylight, when possible; use of swings, climbing handles, and other structures that maximize use of space; training animals to move during enclosure cleaning to avoid wetting; training animals to check exits to minimize intrusions into enclosure	Pens or runs with wood chip substrate and of sufficient height to permit full vertical stretching; front and sidewalls that permit dogs to see other dogs as well as personnel entering and exiting the holding room, use of horizontal bars instead of vertical in gates and walls; elevated resting perch; dog beds; removal of animals from pens during cleaning to avoid wetting	Solid bottom housing well bedded with cellulose- or wood-based substrate; enclosures that provide sufficient height and space for full vertical and horizontal stretching; multi-level housing and/or grid walls for climbing; opportunities to dig; shelter or tunnel and nesting material, such as strips of crinkle paper [99]	Pen housing with solid bottom floor and natural substrate and of a height that permits full vertical standing on haunches (‘periscope’) without ear tips touching enclosure top; shelter (e.g., PVC, cardboard box, large empty paper shavings bags, etc.); elevated resting perch made of thermoneutral material	Pen housing with solid bottom floor and natural substrate (e.g., straw, wood shavings); perforated or plastic side and front walls/gates that permit pigs to see other pigs as well as personnel entering or exiting the holding room; use of horizontal bars [100] instead of vertical in gates and walls; plastic baths (‘beds’); shelf that provides overhead shelter; removal of animals from pens during cleaning to avoid wetting with soiled water or standing on wet floors [101].
Exercise	Regular opportunities for free movement that allow sufficient space for performance of all gaits and natural movements; e.g., running, jumping, hopping, climbing, and exploring. Maintains and improves gross and fine motor control, maintains normal musculature and promotes normal bone density, promotes metabolism [102] and reduces obesity, enhances psychological well-being and learning, permits juvenile and young adult animals to work off pent-up energy and/or to play, and encourages exploration and cognitive engagement. Note: exercise needs may be met within the confines of a well-designed housing system	Exercise pen; kiddie pool [103]; use of climbing handles on pen walls; swings or suspended tunnels or shelves	Indoor or outdoor leash walking; dedicated exercise rooms or temporary use of hallways or enclosure rooms for free running; use of play equipment for small children such as ramps; kiddie pools	Large plastic or wooden tubs or boxes with novel sheltering and climbing structures; painters’ tray filled with water; box with sterilized soil for digging	Large, deep wheeled cart with wood chip substrate; fenced floor pens [104]	Out of enclosure time in holding room or corridor on flooring that minimizes slips and falls; kiddie pools or showers; ‘rooting pen’
Habituation and desensitization	Behavioral techniques to reduce fear to a stimulus following repeated exposure, such as the presence of husbandry or experimental personnel, equipment or sounds, such that the animal learns that the stimulus is not threatening. Habituation is a passive form of learning that better prepares animals for experimental work. Requires additional time pre-study but helps to ensure animals remain unstressed and reduces struggling and potential for injuries	Accepting food treats by hand, desensitization to primate handling gloves, repeated neutral to positive exposure to restraint devices and other experimental equipment and setups that the animal will be in contact with	Personnel sitting on floor of holding room and allowing animal to interact freely with them, repeated neutral to positive exposure to experimental rooms and equipment	Short periods of out-of-enclosure handling and touching using a VetBed or soft fleece, use of small food treats; placing restraint or inhalation tubes in pens to permit exploration and desensitization	Provision of a food treat at front of enclosure, repeated exposure to restraint equipment	Short periods of gentle touching or nearness to person while eating granola or small treats
Animal training	Operant conditioning in which animals are rewarded (e.g., verbal praise, food reward, positive touching, use of manipulable toy or activity board) for responding appropriately to specific cues or commands [105,106]. Requires time to shape behaviors as well as consistency in use of techniques. Promotes voluntary cooperation with techniques, promotes cognitive engagement of animal, and reduces human and animal injuries. Requires additional time pre-study. Social learning is possible and enhances speed of learning for most species [107,108]	Calmly moving out of enclosure into restraint chair, moving into transfer tunnel or box, cooperation with technique to reduce restraint or need for sedation or anesthesia, e.g., bleeding, dosing, swabbing; participating in studies via touchscreen [109,110]	Sitting on platforms quietly for inhalation or infusion dosing, jumping onto exam table, running out of pen or housing room to procedure room [111]	Sitting quietly on lap or in shelter during infusions, injections or blood collection [112]	Sitting quietly in restraint device, on lap or on platform or equipment during dosing, injection, blood collection or an experimental procedure	Walking out of enclosure onto floor scale; walking out of enclosure into elevated treatment area or onto a hoist for dosing [89]
Handling and restraint	Use of nonaversive or low stress handling techniques that minimize fear or distress, bites and scratches, animal injuries, and promote trust; ergonomically sound practices that reduce human injuries. Minimize periods of restraint and consider providing rest periods for longer duration restraint. Animals should be habituated to devices for prolonged (i.e., greater than 15 min) periods of restraint	Use of chairs made of thermoneutral materials that permit comfortable postures and sitting; minimizing use of tie ropes	Gentle manual restraint for short techniques; use of platforms for long-term restraint, slings for shorter periods of restraint; minimizing use of tie ropes	Gentle manual restraint with soft drape for short techniques; use of devices made of thermoneutral materials to which animals have been habituated	Gentle manual restraint and cover eyes for short techniques; for longer periods use of techniques (e.g., bunny burrito) and devices made of thermoneutral materials to which animals have been habituated	Encourage free walking to minimize carrying, lifting and manual restraint; use of platforms, hydraulic lifts, and slings for short periods of restraint; minimize use of tie ropes
Choice and control	Refers to animal having sufficient space and options, including in their environment, and in their ability to exert some degree of control over themselves, e.g., desire to sit, eat or rest near or away from other conspecifics, and their ability to cooperate with various procedures and interactions with humans [113]	Enclosure size and design; volume or on/off control for radio or television, training for cooperation [114,115,116]	Enclosure size and design, training for cooperation	Enclosure size and design, desensitization and training for cooperation	Enclosure size and design, training for cooperation	Enclosure size and design, training for cooperation

## Data Availability

Not applicable.

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
