# Peer review of "Research Animal Behavioral Management Programs for the 21st Century"

_animals, 2023, doi:10.3390/ani13121919_

Round 1

Reviewer 1 Report

This Review aims at presenting an “updated behavioral management program approach” to improve animal welfare and the quality of science.

The review presents The Umbrella Model of Research Animal Behavioral Management Programs, in which the authors have joined all the aspects of the Animal Behavioral Management program.

Overall, many examples are given, but only a few references and explanations. Also, I would have liked a more experienced-based approach to this umbrella-model. The subtle details in animal training (habituation, desensitization, operant training etc.) are ignored. It must be acknowledged that a poor animal trainer will risk adding to stress and frustration as much as a good animal trainer will promote trust, cooperation and increased animal welfare.

Comment no. 1

It is briefly mentioned that The Association of Zoos and Aquariums are using such programs, but actually the entire concept of behavioural management stems from this sector and I think it is vital to address how Zoo and Aquariums have developed this concept over the last 30 years or more

The authors need to consider e.g. the work done by The Animal Behavior Management Alliance (ABMA) – a not-for-profit corporation for animal care professionals working to enhance animal care through training and enrichment. The ABMA has existed since 2000 – and explain how this new “Umbrella Approach” differs from the approach of ABMA. We, who work with laboratory animals, should rather address the constraints that we have when we try to implement this well-known approach.

Comment no. 2

The authors presents both BMP (behavioral Management Program) and their Research Animal Behavioral Management Program (RABMP), but I find it difficult to see the difference and how the RABMP is an improvement of the BMP. In the BMP “facility design” is a part if the definition, but this is not – as far as I can see – included in the Umbrella model. Facility design covers many things, and could include e.g. snout-contact-openings in walls between single-housed pigs or the mindful pacing of the scale in the pig facility. I would like to know why this aspect is not included under the Umbrella.

Choice and Control: In this part of the RABMP, enclosure design is included – however, the concept of “Choice and Control” is also valid to use when discussing animal training. So I would suggest adding a reference or two here to emphasize how this concept is defined and used in the umbrella model. In the cited paper (franks 2019), choice and control is mentioned as being relevant for preference-testing using operant techniques (same as in Shapiro & Lambeth 2010).

Comment no. 3

The author should explain why e.g. zebrafish are not part of table 2, Zebrafish are becoming increasingly popular and should be included in a suggestion for a general, broad model improving animal welfare and scientific outcomes.

Comment no. 4.

Human interactions. Why is this not part of the “habituation and desensitization or the training? The authors should explain the concepts of habituation and desensitization in more details. These are often techniques difficult for many people to understand.

Comment no. 5.

Housing and husbandry – the pig examples. Please explain why horizontal bars are better than vertical bars- Please explain why pigs should be removed from pens during cleaning to avoid wetting. Most pigs, I know, love to play with water…

Comment no. 6

Animal training. The authors need to provide appropriate references for this section. I seriously doubt that most people use animal training to make e.g. rats and rabbits sit calm in a restraint device; more often these animals are habituated or in some way forced to sit there. Using animal training as described would require the animals to voluntarily go into the device on cue and sit calm until cued to leave. Please also provide references on the statement “Social learning … enhances speed of learning for most species.”

Comment no. 7

It is not clear if the RABMP are to be used on individual- or group level. This is important to state clearly. The example of the use of the TINT for groups of mice should be explained in more details. If you have a cage with 8 mice and only one or two mice are not nest-building, but the remaining 6 mice are – how will the TINT show that there is a problem in that cage?

End.

Author Response

Reviewer 1

This Review aims at presenting an “updated behavioral management program approach” to improve animal welfare and the quality of science.

The review presents The Umbrella Model of Research Animal Behavioral Management Programs, in which the authors have joined all the aspects of the Animal Behavioral Management program.

Overall, many examples are given, but only a few references and explanations. Also, I would have liked a more experienced-based approach to this umbrella-model.

Thank-you for this comment. We have expanded explanations in Table 2 and added >20 references to better support the recommendations being made.

The subtle details in animal training (habituation, desensitization, operant training etc.) are ignored. It must be acknowledged that a poor animal trainer will risk adding to stress and frustration as much as a good animal trainer will promote trust, cooperation and increased animal welfare.

We agree with the reviewer and we highlight the benefit of working with an animal behaviorist on pg 14, line 373. In addition, we have added a paragraph about tailoring the program to the capabilities of staff, which addresses the reviewer’s comment.

Comment no. 1

It is briefly mentioned that The Association of Zoos and Aquariums are using such programs, but actually the entire concept of behavioural management stems from this sector and I think it is vital to address how Zoo and Aquariums have developed this concept over the last 30 years or more

Thank-you, this is mentioned on pg. 1 – line lines 42-46 and pgs 2-3 – lines 84-110. Please note that this was not intended to be a review of behavior management programs at zoos

The authors need to consider e.g. the work done by The Animal Behavior Management Alliance (ABMA) – a not-for-profit corporation for animal care professionals working to enhance animal care through training and enrichment. The ABMA has existed since 2000 – and explain how this new “Umbrella Approach” differs from the approach of ABMA. We, who work with laboratory animals, should rather address the constraints that we have when we try to implement this well-known approach.

Thank-you for your comments. We have added a paragraph on pg 5, starting at line 216 to recognize the role that other organizations have played in developing and supporting the concept of a BMP. The RABMP model that we describe in this paper is different and expanded in concept from that described on the ABMA website (under the ABMA Position on Animal Ambassadors):

 “The Animal Behavior Management Alliance fully supports the use of animals as ambassadors regardless of species when, a comprehensive behavior management program is in place. This behavior management program should be driven primarily by the use of positive reinforcement and should combine operant conditioning with goal-based enrichment. In addition, the ABMA believes that all ambassador animals require species appropriate housing, diet, and veterinary care to achieve the highest level of welfare.”

Many of the specifics noted and referenced in the current paper and model are different and/or expanded upon.

Comment no. 2

The authors presents both BMP (behavioral Management Program) and their Research Animal Behavioral Management Program (RABMP), but I find it difficult to see the difference and how the RABMP is an improvement of the BMP. In the BMP “facility design” is a part if the definition, but this is not – as far as I can see – included in the Umbrella model. Facility design covers many things, and could include e.g. snout-contact-openings in walls between single-housed pigs or the mindful pacing of the scale in the pig facility. I would like to know why this aspect is not included under the Umbrella.

Thank-you for your comment. We have expanded the text discussion of ‘housing’ on pg 8 to clarify our intent of incorporating ‘facility design’, which we agree is crucial. Aspects of facility design may also be covered under ‘comfort’ and ‘choice and control’.

Choice and Control: In this part of the RABMP, enclosure design is included – however, the concept of “Choice and Control” is also valid to use when discussing animal training. So I would suggest adding a reference or two here to emphasize how this concept is defined and used in the umbrella model. In the cited paper (franks 2019), choice and control is mentioned as being relevant for preference-testing using operant techniques (same as in Shapiro & Lambeth 2010).

Thank-you, great comment and we agree! We have added additional details and references in the table to address the comment.

Comment no. 3

The author should explain why e.g. zebrafish are not part of table 2, Zebrafish are becoming increasingly popular and should be included in a suggestion for a general, broad model improving animal welfare and scientific outcomes.

Thank-you for your comment. Table 2 is not intended to be exhaustive and include every species – for example, we chose to highlight rats instead of mice, and there were fewer references available for zebrafish. To address this gap, we have added new verbiage and additional references beginning on line 397 on pg 15.

Comment no. 4.

Human interactions. Why is this not part of the “habituation and desensitization or the training? The authors should explain the concepts of habituation and desensitization in more details. These are often techniques difficult for many people to understand.

Thank-you, we did define these in Table 2., column 2. An additional reference has been added as well as additional examples. Habituation and desensitization are specific behavioral techniques; not incidental interactions with staff centered on husbandry or research procedures. While we agree that human-animal interactions can be a form of desensitization, it is an activity that can and should occur any time and throughout the duration of the animal’s time in the facility.

Comment no. 5.

Housing and husbandry – the pig examples. Please explain why horizontal bars are better than vertical bars- Please explain why pigs should be removed from pens during cleaning to avoid wetting. Most pigs, I know, love to play with water…

Added several references, as requested. Clarified that it is prolonged standing in water or with wet feet that can be problematic (i.e., leads to fungal infections)

Comment no. 6

Animal training. The authors need to provide appropriate references for this section. I seriously doubt that most people use animal training to make e.g. rats and rabbits sit calm in a restraint device; more often these animals are habituated or in some way forced to sit there. Using animal training as described would require the animals to voluntarily go into the device on cue and sit calm until cued to leave. Please also provide references on the statement “Social learning … enhances speed of learning for most species.”

That is precisely what we mean and it has been employed successfully by many researchers as well as by at least one GLP CRO in Sweden. Several specific references have been added. Regarding social learning, additional references have been added.

Comment no. 7

It is not clear if the RABMP are to be used on individual- or group level. This is important to state clearly. The example of the use of the TINT for groups of mice should be explained in more details. If you have a cage with 8 mice and only one or two mice are not nest-building, but the remaining 6 mice are – how will the TINT show that there is a problem in that cage?

Addressed on pg. 5, lines 224-225.The program can be applied to an individual or a group and both require consideration. We establish programs what on what is known for a species and then tailor to individual response, when possible.

Reviewer 2 Report

A nice summary of the current situation and possible outcomes. I find the article very well done and the first time in my reviewer time no significant deficiencies to correct. Only the introduction could have been a bit better researched with a higher number of citations. But this is only a small comment. Very nice article ... Congratulations

Very good to read. The grammatical aspect is difficult for me to control, as I am not a native speaker of English.

Author Response

Thank-you for your positive comments. We have corrected several typos in the MS.

Reviewer 3 Report

In this review, the authors recall the recent progress in our understanding of animal sentience. The consideration of animal welfare has gradually improved over the years, and many laboratories and institutes around the world have put in place increasingly effective tools and methods to address it.

Sensitive to ethical and regulatory requirements such as Directive 2010/63, the research community has endeavored to ensure basic care by refining the housing conditions of the animals. As noted in the introduction, some species held in animal facilities do not fully benefit from advances in this area.

The introduction sets the context very well, and in the second part, the authors provide a historical review of the development of behavior management programs. This historical perspective is an opportunity to point out that these programs originated in the zoo community, where most animals are housed for very long periods of time and can therefore benefit from long-term monitoring.

The notion of duration is crucial here and deserves to be substantiated. Animals held for very long periods of time will benefit more from behavioral management programs than short-lived animals or animals held for shorter periods of time. This implies a significant effort from scientists not to weigh the length of the animal's life in the laboratory against the investment to ensure its welfare. The public would not accept, for example, that the acclimation time of an animal is reduced simply because it is involved in a terminal procedure for the purpose of organ and tissue removal. From a scientific point of view, the validity of results obtained from an animal that has not sufficiently recovered from its transfer to the animal facility would be questionable. From an ethical point of view, it would not be acceptable not to give the necessary rest to an animal.

This second section also states that one of the barriers to the widespread use of management programs lies in their definition. The very term "enrichment" denotes a historical lack of consideration for the environmental conditions of the animals, and inevitably suggests very impoverished conditions. It then became convenient to confuse enrichment with addition, and to emphasize the introduction of objects, seeking an immediate benefit on the quality of life of the animal held, without precisely evaluating the impact on its well-being.

The definition of enrichment has gradually shifted to focus on its purpose, such as promoting the widest possible repertoire of behaviors. Thus, there remains a utilitarian notion of enrichment, which is more in line with refinement of housing than with a true consideration of animal well-being. This is probably because there are as many definitions of well-being as there are scientists.

This second part also highlights the idea that behavioral management programs have mainly concerned primates. A phylogenic proximity between humans and non-human primates has indeed facilitated this. The concern for primates’ physical well-being was quickly joined by a concern for their psychological health and consequently the idea of providing them with an environment conducive to personal development.

It is also clear from this section that the notion of familiarity with species such as dogs and primates has contributed to the strengthening of large animal programs and has generated public interest. One could also add the notion of affective attachment between humans and animals, an attachment that is much easier with animals that are close to us or that have lived with us for a long time, than with more discreet species such as rats and mice (although their ecosystem overlaps with ours). Familiarity and human attachment to certain species both facilitate the evaluation of the benefits of enrichment programs on well-being.

This historical perspective also recalls the creation of the NIH management plan for non-human primates, an opportunity for the authors to introduce a definition of a behavioral management program that integrates several components, environmental enrichment, social housing, welfare assessment tools and methodologies, and the impact of routine animal operations. Interestingly, this plan also promoted the importance of training/conditioning animals to cooperate. This notion of cooperation would also be interesting to elaborate on. If it is relatively easy to observe an animal cooperating, can we say that it is fulfilling its own expectation, or that of the human handling it?

This review expresses the desire to see animal behavior management programs extended to species other than non-human primates or carnivores. The generalization of these programs to all species is perfectly justified and commendable in view of regulatory, scientific, and societal expectations. One could also question the relevance of such programs or their adaptation to the context of animal research conducted in the field.

Throughout the manuscript, the authors mention several elements that may appear to be obstacles to the development of such programs:

First, there is an economic barrier, because the implementation of a management program may require the acquisition of new materials and equipment, the recruitment of dedicated staff or the consultation of experts in ethology when this expertise is not available locally. Currently, many rodent facilities provide wooden sticks, nesting materials, sometimes tunnels or shelter, and rarely more. This is certainly an improvement over the previous situation of an impoverished environment, but in terms of meeting ethical and societal expectations about lab animals’ well-being, is it only close to the optimum?

A second, but important, obstacle is a scientific one. Numerous studies have shown significant differences between results obtained from animals kept under old standard conditions and from animals that benefit from even minimal environmental enrichment. This can question the scientific validity of previous studies, but also the need to repeat certain studies in a context that is now more favorable to animals. As an example, Gouveia and Hurst show that manipulating mice with tunnels not only reduces their anxiety (doi:10.1371/journal.pone.006640) but also results in better performance in behavioral tests (DOI: 10.1038/srep44999). The question that emerges is, what have we previously shown on our animals in a depleted environment? Given that, should we not fear some reluctance from the scientific community to implement behavior management programs?

There is also a human limitation, because although we may not like it, we are also concerned about our own comfort and environmental safety. Animal caretakers also have expectations that need to be listened to. Just as the assessment of animal welfare depends on the welfare of the professionals, the effectiveness of a behavior management program will inevitably depend on the involvement of the staff. It might therefore be interesting to add to the twelve items, a separate consideration for animal staff management. The outcome could be « Being aware of our own needs and being able to express them, in order to facilitate understanding of the animals’ individual needs ». This would be in phase with the “One health” concept.

Taking care of animals is a job and like any job, requires efforts. Significant progress has been made in easing the burden of caretakers tasks, whether it be for bedding change, cages handling and facilities management. But have we sufficiently assessed the impact of these advances and routine animal care operations on animals’ well-being? What about the impact of changing animals under a noisy hood and bright lights, or of having food available ad libitum? Many of us have observed a rapid weight gain in laboratory rodents. There are clear links between food availability, stress, boredom, lack of physical activity and weight gain. However, are we willing or ready to change our practices and stop feeding animals at will and opt for a diet more in line with the needs and rhythms of their species?

The human factor cannot be discussed without mentioning consideration for the animal. While there is unanimity on the need to ensure animal welfare, there is a significant difference in its perception between the scientific community and the public. This is particularly well illustrated by the study of Franco et al (2018), which clearly shows a reversal of hierarchy of the 3Rs between societal expectations and those of scientists. This translates into a predominance of concerns for refinement and pain prevention, naturally both important. In fact, we have all seen a clear improvement in animal care over the last years, while the public and society expect a better consideration for replacement and, when not possible, for the well-being of animals. In this respect, the European Citizens' Initiative "Save cruelty free cosmetics - commit to a Europe without animal testing" (Commission registration number: ECI(2021)000006) is perhaps an early sign of a breakup between the scientific community and the public. The scientific community should react promptly, to not jeopardize the possibility of using animals for research and teaching.

The proposed framework is in line with a refinement approach. As the authors rightly state, such programs can improve the fate of our animals, but still they may not fully meet ethical and societal expectations.

This is particularly well illustrated by enrichment, whose concept has gradually turned into a utilitarian and reductionist issue. An enrichment program may indeed contribute to the reduction of abnormal behavior in animals. However, does the absence of abnormal behavior mean that the animals are in a state of well-being? If environmental enrichment improves the quality of human-animal interactions, can we really say that the animal is alright in its body and mind? How can we ensure that a cooperative animal is truly in a state of well-being?

Looking at Figure 1, the authors’ definition of behavioral management program clearly appears holistic and implies a comprehensive consideration of each of the twelve items under the umbrella. This is ambitious and, to some extent, idealistic.

The table provides examples for each of the items for several species, with an outcome-based approach. Several examples are interesting but also illustrative of the limitations of the proposed program. For instance, things to chew are provided to reduce boredom and to satisfy the need for chewing for animals with continuously growing teeth. The observation of animals chewing will certainly confirm that they express a behavior specific to their species. However, will this behavior observed at a given moment be representative of an absence of boredom and of a real state of well-being? How can we be sure that an animal that is chewing or biting a piece of wood is psychologically well and is not reacting to stressful factors such as promiscuity, a previous conflict with a congener, a high density of animals in the cage, a ventilation airflow or noise level, or the entrance of a zootechnician in the room?

Regarding behavioral assessment, the program also notes its limitations with the mention of periodic or ‘for cause’. We all want animals that are well acclimatized to their environment and that express their behavior to the best of their ability. However, are we willing and equipped to perform a complete ethogram on an individual basis whenever necessary? Depending on the number of animals housed, especially rodents or fishes, such evaluations are not feasible, and for rodents the authors propose to use the measurement of time to reach the nest, or a nest quality test. The question arises as to whether these tests are truly indicative of the animal's well-being or simply the presence of species-specific behavior.

The definition of animal’s well-being proposed in France by the Agence nationale de sécurité sanitaire de l'alimentation, de l'environnement et du travail (ANSES, ie National Agency for Food, Environmental and Occupational Health Safety) has been enriched by the notions of the animal's expectations and animal’s perception of the situation: “An animal's well-being is the positive mental and physical state associated with the satisfaction of its physiological and behavioral needs and expectations. This state varies according to the animal's perception of the situation.”

In the present framework, one can ask where the animal’s expectations are, and the tools to address them. The animal’s perception of its own situation is dependent on the means the animal is provided with to adjust to its environment. This means that a special consideration is needed for housing conditions, as well as efforts towards the development of more naturalistic and ecologically relevant environments. This is a real challenge because little environmental complexity in animal housing has long contributed to reduce bias and non-specific effects in animal studies. Now, we know that it was to the detriment of well-being. Introducing complexity in housing and husbandry might therefore increase inter-individual variability, but also give the opportunity to address individual experience and how each animal copes with its environment. It should also increase the representativeness of animals in studies for a better transposition of results to the “real world”.

Considering individuals and their expectations will require new tools to monitor individual behavioral repertoire and experience, including interactions with congeners. Video recordings coupled with artificial intelligence programs could in this matter be extremely valuable and also a subject for future research.

In conclusion, I would like to thank the authors for this particularly enlightening and inspiring review. Collectively we can only hope that behavioral management programs will be gradually implemented and generalized, as it will contribute to the refinement of animal housing and care and, consequently, to the scientific quality of animal studies. The scientific community, through its commitment, publications, and communications, must also give signs to the general public of its efforts for the 3Rs and for a better consideration of the well-being of each lab animal.

Author Response

In this review, the authors recall the recent progress in our understanding of animal sentience. The consideration of animal welfare has gradually improved over the years, and many laboratories and institutes around the world have put in place increasingly effective tools and methods to address it.

Sensitive to ethical and regulatory requirements such as Directive 2010/63, the research community has endeavored to ensure basic care by refining the housing conditions of the animals. As noted in the introduction, some species held in animal facilities do not fully benefit from advances in this area.

The introduction sets the context very well, and in the second part, the authors provide a historical review of the development of behavior management programs. This historical perspective is an opportunity to point out that these programs originated in the zoo community, where most animals are housed for very long periods of time and can therefore benefit from long-term monitoring.

The notion of duration is crucial here and deserves to be substantiated. Animals held for very long periods of time will benefit more from behavioral management programs than short-lived animals or animals held for shorter periods of time. This implies a significant effort from scientists not to weigh the length of the animal's life in the laboratory against the investment to ensure its welfare. The public would not accept, for example, that the acclimation time of an animal is reduced simply because it is involved in a terminal procedure for the purpose of organ and tissue removal. From a scientific point of view, the validity of results obtained from an animal that has not sufficiently recovered from its transfer to the animal facility would be questionable. From an ethical point of view, it would not be acceptable not to give the necessary rest to an animal.

Thank-you, we have added a notation and reference to emphasize this comment per the reviewer’s suggestion.

This second section also states that one of the barriers to the widespread use of management programs lies in their definition. The very term "enrichment" denotes a historical lack of consideration for the environmental conditions of the animals, and inevitably suggests very impoverished conditions. It then became convenient to confuse enrichment with addition, and to emphasize the introduction of objects, seeking an immediate benefit on the quality of life of the animal held, without precisely evaluating the impact on its well-being.

The definition of enrichment has gradually shifted to focus on its purpose, such as promoting the widest possible repertoire of behaviors. Thus, there remains a utilitarian notion of enrichment, which is more in line with refinement of housing than with a true consideration of animal well-being. This is probably because there are as many definitions of well-being as there are scientists.

This second part also highlights the idea that behavioral management programs have mainly concerned primates. A phylogenic proximity between humans and non-human primates has indeed facilitated this. The concern for primates’ physical well-being was quickly joined by a concern for their psychological health and consequently the idea of providing them with an environment conducive to personal development.

It is also clear from this section that the notion of familiarity with species such as dogs and primates has contributed to the strengthening of large animal programs and has generated public interest. One could also add the notion of affective attachment between humans and animals, an attachment that is much easier with animals that are close to us or that have lived with us for a long time, than with more discreet species such as rats and mice (although their ecosystem overlaps with ours). Familiarity and human attachment to certain species both facilitate the evaluation of the benefits of enrichment programs on well-being.

Thank-you for this comment. A notation and reference have been added to pg 2 to emphasize this point.

This historical perspective also recalls the creation of the NIH management plan for non-human primates, an opportunity for the authors to introduce a definition of a behavioral management program that integrates several components, environmental enrichment, social housing, welfare assessment tools and methodologies, and the impact of routine animal operations. Interestingly, this plan also promoted the importance of training/conditioning animals to cooperate. This notion of cooperation would also be interesting to elaborate on. If it is relatively easy to observe an animal cooperating, can we say that it is fulfilling its own expectation, or that of the human handling it?

An interesting point. More likely the human expectation, but in so doing, creates a beneficial effect of reducing self stress.

This review expresses the desire to see animal behavior management programs extended to species other than non-human primates or carnivores. The generalization of these programs to all species is perfectly justified and commendable in view of regulatory, scientific, and societal expectations. One could also question the relevance of such programs or their adaptation to the context of animal research conducted in the field.

Throughout the manuscript, the authors mention several elements that may appear to be obstacles to the development of such programs:

First, there is an economic barrier, because the implementation of a management program may require the acquisition of new materials and equipment, the recruitment of dedicated staff or the consultation of experts in ethology when this expertise is not available locally. Currently, many rodent facilities provide wooden sticks, nesting materials, sometimes tunnels or shelter, and rarely more. This is certainly an improvement over the previous situation of an impoverished environment, but in terms of meeting ethical and societal expectations about lab animals’ well-being, is it only close to the optimum?

A second, but important, obstacle is a scientific one. Numerous studies have shown significant differences between results obtained from animals kept under old standard conditions and from animals that benefit from even minimal environmental enrichment. This can question the scientific validity of previous studies, but also the need to repeat certain studies in a context that is now more favorable to animals. As an example, Gouveia and Hurst show that manipulating mice with tunnels not only reduces their anxiety (doi:10.1371/journal.pone.006640) but also results in better performance in behavioral tests (DOI: 10.1038/srep44999). The question that emerges is, what have we previously shown on our animals in a depleted environment? Given that, should we not fear some reluctance from the scientific community to implement behavior management programs?

Thank-you, we discuss this on pg 6 and provide several references that question the results obtained using conventionally housed mice. There is a significant concern that not attending to behavioral needs results in less translatable models (unless we are specifically hoping to study inactive, frustrated, and obese humans, for example).

There is also a human limitation, because although we may not like it, we are also concerned about our own comfort and environmental safety. Animal caretakers also have expectations that need to be listened to. Just as the assessment of animal welfare depends on the welfare of the professionals, the effectiveness of a behavior management program will inevitably depend on the involvement of the staff. It might therefore be interesting to add to the twelve items, a separate consideration for animal staff management. The outcome could be « Being aware of our own needs and being able to express them, in order to facilitate understanding of the animals’ individual needs ». This would be in phase with the “One health” concept.

Thank-you, we have added a sentence to this effect at line 368.

Taking care of animals is a job and like any job, requires efforts. Significant progress has been made in easing the burden of caretakers tasks, whether it be for bedding change, cages handling and facilities management. But have we sufficiently assessed the impact of these advances and routine animal care operations on animals’ well-being? What about the impact of changing animals under a noisy hood and bright lights, or of having food available ad libitum? Many of us have observed a rapid weight gain in laboratory rodents. There are clear links between food availability, stress, boredom, lack of physical activity and weight gain. However, are we willing or ready to change our practices and stop feeding animals at will and opt for a diet more in line with the needs and rhythms of their species?

The human factor cannot be discussed without mentioning consideration for the animal. While there is unanimity on the need to ensure animal welfare, there is a significant difference in its perception between the scientific community and the public. This is particularly well illustrated by the study of Franco et al (2018), which clearly shows a reversal of hierarchy of the 3Rs between societal expectations and those of scientists. This translates into a predominance of concerns for refinement and pain prevention, naturally both important. In fact, we have all seen a clear improvement in animal care over the last years, while the public and society expect a better consideration for replacement and, when not possible, for the well-being of animals. In this respect, the European Citizens' Initiative "Save cruelty free cosmetics - commit to a Europe without animal testing" (Commission registration number: ECI(2021)000006) is perhaps an early sign of a breakup between the scientific community and the public. The scientific community should react promptly, to not jeopardize the possibility of using animals for research and teaching.

We have added another sentence on change management and institutional culture to emphasize this point.

The proposed framework is in line with a refinement approach. As the authors rightly state, such programs can improve the fate of our animals, but still they may not fully meet ethical and societal expectations.

This is particularly well illustrated by enrichment, whose concept has gradually turned into a utilitarian and reductionist issue. An enrichment program may indeed contribute to the reduction of abnormal behavior in animals. However, does the absence of abnormal behavior mean that the animals are in a state of well-being? If environmental enrichment improves the quality of human-animal interactions, can we really say that the animal is alright in its body and mind? How can we ensure that a cooperative animal is truly in a state of well-being?

Looking at Figure 1, the authors’ definition of behavioral management program clearly appears holistic and implies a comprehensive consideration of each of the twelve items under the umbrella. This is ambitious and, to some extent, idealistic.

The table provides examples for each of the items for several species, with an outcome-based approach. Several examples are interesting but also illustrative of the limitations of the proposed program. For instance, things to chew are provided to reduce boredom and to satisfy the need for chewing for animals with continuously growing teeth. The observation of animals chewing will certainly confirm that they express a behavior specific to their species. However, will this behavior observed at a given moment be representative of an absence of boredom and of a real state of well-being? How can we be sure that an animal that is chewing or biting a piece of wood is psychologically well and is not reacting to stressful factors such as promiscuity, a previous conflict with a congener, a high density of animals in the cage, a ventilation airflow or noise level, or the entrance of a zootechnician in the room? We can not be certain of motivation, of course, and can only try to relate behaviors with a holistic picture of overall animal well-being. We cannot recapitulate a free-living state within a vivarium and that is not the intent

Thank-you - a comment has been added to the end of pg 8 to this effect..

Regarding behavioral assessment, the program also notes its limitations with the mention of periodic or ‘for cause’. We all want animals that are well acclimatized to their environment and that express their behavior to the best of their ability. However, are we willing and equipped to perform a complete ethogram on an individual basis whenever necessary? Depending on the number of animals housed, especially rodents or fishes, such evaluations are not feasible, and for rodents the authors propose to use the measurement of time to reach the nest, or a nest quality test. The question arises as to whether these tests are truly indicative of the animal's well-being or simply the presence of species-specific behavior.

A notation has been added to this section to indicate that this is a surrogate measure of mouse welfare.

The definition of animal’s well-being proposed in France by the Agence nationale de sécurité sanitaire de l'alimentation, de l'environnement et du travail (ANSES, ie National Agency for Food, Environmental and Occupational Health Safety) has been enriched by the notions of the animal's expectations and animal’s perception of the situation: “An animal's well-being is the positive mental and physical state associated with the satisfaction of its physiological and behavioral needs and expectations. This state varies according to the animal's perception of the situation.”

In the present framework, one can ask where the animal’s expectations are, and the tools to address them. The animal’s perception of its own situation is dependent on the means the animal is provided with to adjust to its environment. This means that a special consideration is needed for housing conditions, as well as efforts towards the development of more naturalistic and ecologically relevant environments. This is a real challenge because little environmental complexity in animal housing has long contributed to reduce bias and non-specific effects in animal studies. Now, we know that it was to the detriment of well-being. Introducing complexity in housing and husbandry might therefore increase inter-individual variability, but also give the opportunity to address individual experience and how each animal copes with its environment. It should also increase the representativeness of animals in studies for a better transposition of results to the “real world”.

The authors strongly concur!

Considering individuals and their expectations will require new tools to monitor individual behavioral repertoire and experience, including interactions with congeners. Video recordings coupled with artificial intelligence programs could in this matter be extremely valuable and also a subject for future research.

Thank-you for this comment, we have added several sentences and 2 references to emphasize this point with zebrafish as an example.

In conclusion, I would like to thank the authors for this particularly enlightening and inspiring review. Collectively we can only hope that behavioral management programs will be gradually implemented and generalized, as it will contribute to the refinement of animal housing and care and, consequently, to the scientific quality of animal studies. The scientific community, through its commitment, publications, and communications, must also give signs to the general public of its efforts for the 3Rs and for a better consideration of the well-being of each lab animal.

Thank-you for your thoughtful comments!